# Indicators of Obesity and Cardiorespiratory Fitness in Croatian Children

**DOI:** 10.3390/jfmk9040250

**Published:** 2024-11-29

**Authors:** Marko Badrić, Leona Roca, Vladan Pelemiš, Dragan Branković, Vladimir Živanović

**Affiliations:** 1Faculty of Teacher Education, University of Zagreb, 10000 Zagreb, Croatia; marko.badric@ufzg.hr (M.B.); leona.roca@ufzg.hr (L.R.); 2Faculty of Education, University of Belgrade, 11000 Belgrade, Serbiadragan.brankovic@uf.bg.ac.rs (D.B.)

**Keywords:** cardiorespiratory fitness, obesity, students, body composition, physical exercise

## Abstract

Background/Objectives: The aim of this study is to investigate the relationship between CRF (cardiorespiratory fitness) and body composition, specifically obesity indicators, in a sample of schoolchildren from the continental and Adriatic regions of the Republic of Croatia. Given that Croatia ranks among the leading countries in Europe in terms of obesity, it is believed that there is a need to combat this condition starting from school age. The usefulness of cardiorespiratory fitness (CRF) lies in the fact that it is one of the indicators of children’s overall health and is considered a measure of general health status. The findings will emphasize the need to determine CRF and BMI as important factors that should be addressed from early school years in order to prevent future cardiovascular diseases. Methods: The total sample consisted of 423 students, including 212 girls and 211 boys, from the continental and Adriatic regions of the Republic of Croatia, with an average age of 10.32 years. Body height was measured using a portable stadiometer, while body weight, body mass index (BMI), and body fat percentage were measured using a dual-frequency body composition analyzer (TANITA DC-360P) based on the principles of bioelectrical impedance. The WHR index was calculated as the ratio of waist to hips, while the WHtR index was calculated as the ratio of waist circumference to height. Cardiorespiratory fitness was assessed using the 20-meter multistage shuttle run test. Results: The results show that 25% of the participants were overweight, and 9% were obese. The ANOVA results indicate that the boys had significantly higher values for BMI, WC, HC, WHtR, and WHR, while the girls had a significantly higher body fat percentage (BF%). Additionally, the boys had better CRF, as they ran more meters in the 20-meter shuttle run test. The variance analysis revealed that the participants with normal body weight had significantly higher cardiorespiratory capacity values compared to overweight or obese students. Regression analysis showed statistical significance for the set of predictor variables, which were BMI indicators, on the criterion variable predicting cardiorespiratory fitness (CRF) in both genders. The coefficient of determination (R^2^) explained almost 22% of the shared variability in boys and 19% in girls. Statistically significant beta coefficients were found for body fat percentage (−0.60) in boys and (−0.98) in girls. Conclusions: Body fat percentage (BF%) was shown to be a significant and useful predictor of cardiorespiratory fitness in school-aged boys and girls. Maintaining an optimal body weight along with high levels of CRF should be an important goal in promoting health among children and adolescents.

## 1. Introduction

In 2022, more than 390 million children and adolescents aged 5 to 19 were overweight, including 160 million who were obese. The prevalence of overweight (including obesity) among children and adolescents aged 5 to 19 has dramatically increased from just 8% in 1990 to 20% in 2022 [1]. In a study conducted in the Republic of Croatia in the 2021/2022 school year, 36.1% of children aged 8 to 8.9 years were found to be overweight or obese, with 38.5% of boys and 33.7% of girls affected [2]. Compared to the same study conducted in 2018/2019, this represented a 3.14% increase in the number of overweight or obese students aged between 8 and 9 years. These increasing trends indicate that, on average, the number of overweight or obese students in the Republic of Croatia grows by 1% annually, showing that the recommendations of the World Health Organization to reduce childhood obesity are not being followed. School children from the Republic of Croatia rank among the top five European countries for overweight and obesity rates (WHO, 2022) [3]. One factor that can directly influence the reduction in obesity in children is cardiorespiratory fitness. Cardiorespiratory fitness (CRF) is a component of physical fitness that reflects the integrated function of numerous bodily systems involved in the circulation and utilization of oxygen to support muscle activity during continuous whole-body physical activity [4]. CRF reflects the fitness of the oxygen transport system to deliver oxygen to muscles when performing physical work, and more oxygen is required to complete this work [5]. Previous research clearly shows that cardiorespiratory fitness is an indicator of overall health and is significantly associated with health outcomes such as obesity and abdominal obesity in youth [6,7]. CRF in children and young people has long been recognized as an important factor for measurement and monitoring, with CRF being significantly linked to health and, independently of physical activity levels, an important determinant in sports [8]. The utility of CRF lies in its status as one of the key indicators of overall child health [8,9], and it is directly linked to the integrated functions of numerous body systems [10], making it the best indicator of health status due to its direct connection to the respiratory and musculoskeletal systems [11]. Almost half of the variance in CRF can be attributed to genetics, especially in the early stages of life, while the rest is primarily modified through regular physical activity [12,13]. Indeed, low CRF in childhood has been recognized as a risk factor for cardiovascular diseases (CVDs) [14]. Cardiorespiratory fitness (CRF) is an important clinical parameter for diagnosing and monitoring the current and future functional and metabolic health of obese young people [15]. It has been proven that better cardiorespiratory fitness is negatively correlated with “metabolic syndrome”, obesity, and diabetes [16]. Moreover, it has been shown that a high level of CRF is strongly associated with a lower risk of premature mortality and the occurrence of chronic conditions (hypertension, heart failure, stroke, atrial fibrillation, dementia, and depression) [17]. This is particularly problematic as obese children have an increased risk of developing cardiovascular diseases [18]. Therefore, there is an urgent need to identify children with obesity and cardiovascular risk factors to implement appropriate health care measures for them [19].

Given that cardiorespiratory fitness (CRF) has been established as a key element of all components of physical fitness in children and adolescents [20], that CRF assessments can be obtained in a pragmatic way, and that CRF can be improved through physical exercise [21], it cannot be maintained or increased without physical activity [13]. As the rising prevalence of obesity and declining physical health in children and adolescents have become major societal issues [22], obesity indicated by body mass index (BMI), waist circumference, body fat percentage, and other measures is a key parameter linking CRF to cardiometabolic risk factors [23,24]. The connection between higher obesity and lower levels of CRF is strongest when adiposity is measured using imaging techniques, somewhat weaker when assessed by waist circumference (WC) and the waist-to-height ratio (WHtR), and weakest when using the body mass index (BMI) [25].

It is a proven fact that obesity is an independent risk factor for cardiovascular diseases, and this condition can be diagnosed in several ways. One of the most common methods is the body mass index (BMI), which is used to classify obesity in children. However, this indicator has limitations since it measures excess body weight rather than excess body fat or fat distribution, and the data obtained are far from ideal, requiring additional information about the impacts which increased obesity has on an individual [26,27]. In contrast to the body mass index (BMI), waist circumference (WC) is an inexpensive, simple, and valid tool [28,29] that has proven to be an effective indicator of abdominal obesity [30], which reflects the accumulation of fat in the abdominal area and is a valid predictor of future cardiometabolic and chronic diseases [31].

Additionally, the waist-to-height ratio (WHtR) is another diagnostic tool that is considered to assess obesity more accurately than BMI, and it is not affected by age and gender [32]. It is a very simple anthropometric index with strong screening power and quick interpretation for the early detection of abdominal obesity in childhood [33]. Previous studies have shown that WHtR is more sensitive, cheaper, and easier to measure and calculate than BMI and WC, and it can be used for both genders [34], with only slight changes as one ages [35]. Moreover, it has been shown that waist circumference, along with body fat percentage (BF%) and WHtR, is a useful predictor of CRF and that somatic indices, excluding WHR, are stronger predictors of CRF than BMI, confirming the assertion that BMI should not be the only index used [36].

A better understanding of the relationship between cardiorespiratory fitness, indicators of obesity or excess body fat, and early cardiometabolic risk factors offers opportunities to promote health even among school-aged children. The aim of this study is to examine the relationship between CRF and body composition, specifically obesity indicators, in a sample of schoolchildren from the continental and Adriatic regions of the Republic of Croatia. The findings will focus on the need to identify CRF and obesity as important factors to address from early school days to prevent future cardiovascular diseases.

## 2. Materials and Methods

### 2.1. Research Procedures and Sample of Participants

Schoolchildren aged 10 to 11 (mean age M = 10.32 and SD = 0.48) from the continental and coastal areas of the Republic of Croatia participated in this cross-sectional study. The selection of elementary schools was carried out using a random sampling method. School principals were contacted by phone or email prior to the study’s implementation, and approval was requested for students’ participation in the research. After obtaining the approval, written consent forms were distributed to students for participation in this study. The research included students who had obtained consent from their parents or guardians and those who independently signed the informational text for students. By signing the informational text, it was assumed that the student agreed to participate in the research. The inclusion criteria for this study were students who were completely healthy without physical or mental impairments, regularly attended physical education classes, had signed parental consent, and completed measurements in all morphological and functional tests that were used for further analyses. This study included fourth-grade students from 10 elementary schools. The total sample consisted of 423 students, of whom 212 were girls and 211 were boys. All procedures conducted in this study were anonymous and were carried out according to the Helsinki Declaration. This research was approved by the Ethics Committee of the Faculty of Education and Rehabilitation Sciences at the University of Zagreb.

### 2.2. Anthropometric Measurements

Body height was measured using a portable stadiometer with a precision of 0.1 cm (Seca^®^ 213, Hamburg, Germany). Body mass, body mass index (BMI), and body fat percentage were measured using a dual-frequency body composition analyzer (TANITA DC-360P: Tanita Corporation, Tokyo, Japan) based on the principles of bioelectrical impedance. Based on the calculated body mass index (BMI), expressed as the ratio of body mass to the square of body height, the participants were classified into three groups according to their nutritional status: normal weight, overweight, and obese, according to reference tables recommended by the International Obesity Task Force [37]. Waist and hip circumferences were measured using a measuring tape. Waist circumference was measured at the midpoint between the lowest rib and the upper boundary of the iliac crest at the end of a normal expiration [38]. Hip circumference was measured at the widest part of the hip, at the level of the greater trochanter. The waist-to-hip ratio (WHR) was calculated based on these measurements. The waist-to-height ratio (WHtR) was calculated as the ratio of waist circumference to height.

### 2.3. Measurement of Cardiorespiratory Fitness

Cardiorespiratory fitness was evaluated using the 20-meter shuttle run test. In this field test, the participants ran back and forth between two marked lines set 20 m apart. The 20-meter distance was indicated by measuring tape and cones. The running speed was controlled by audio signals, starting at 8.5 km/h and increasing by 0.5 km/h every minute. Each level lasted approximately 60 s, with the “speed” (the duration of each interval) dictated to the participant by the sound intervals. The task was for the participant to maintain the designated running rhythm for as long as possible. The final test result was the number of complete 20-meter laps run [39]. Measurement was stopped when a participant failed to maintain the required rhythm twice in a row or voluntarily withdrew due to an inability to continue running. Testing of the students was conducted in school sports halls.

### 2.4. Statistical Analyses

Regarding the descriptive statistics, the following basic statistical parameters were used: mean value (AM), standard deviation (SD), minimum (Min), and maximum (Max) values, as well as kurtosis (Kurt) and skewness (Skew) of the distribution. To determine the relationship between obesity indicators and the results in cardiorespiratory fitness, expressed as the number of laps completed, multivariate (multiple) regression analysis was applied. Differences between genders regarding obesity indicators and cardiorespiratory fitness were examined using a one-way analysis of variance (ANOVA). Pearson’s correlation coefficient was used to assess the individual relationships between variables representing obesity indicators and cardiorespiratory fitness. Statistical significance was tested at a level of *p* < 0.05. All analyses were performed using STATISTICA version 14.0.0.15, TIBCO Software Inc. (Palo Alto, CA, USA).

## 3. Results

Descriptive parameters of the tested participants concerning obesity indicators and results of the CRF test (number of laps) for the total sample are presented in Table 1. The data refer to a group of 423 students aged 10.32 ± 0.48 years. The participants were classified based on their BMI values [37]. Among the tested participants, 25% had excess body weight, and 9% were classified as obese. Observing the results by gender, it is evident that 4% of the girls were obese, while the percentage was 14% for boys. In the group with excess body weight, the results were nearly identical, with 24% of boys compared to 25% of girls.

Table 2 presents the basic descriptive parameters of the sample subjects by gender. The subjects differed significantly in age but not in height and weight. The boys were slightly older and had a higher body mass. When examining the individual differences in the variables used to assess body composition, it is evident that the girls had a significantly higher body fat percentage than the boys and a significantly lower body mass index. Furthermore, the girls had significantly lower values for waist circumference (WC), waist-to-hip ratio (WHR), and waist-to-height ratio (WHtR). The boys statistically outperformed the girls in the 20-meter shuttle run test, with the boys averaging five laps more than the girls. Therefore, we assumed that the boys had a higher level of maximal oxygen uptake (VO_2_max). Compared to the percentile norms established by Tomkinson et al. [40], cardiorespiratory fitness was very low for both the boys and the girls.

The results of the variance analysis in Table 3 show differences in cardiorespiratory fitness according to the degree of nutrition for both genders. It is evident that there were statistically significant differences between the groups categorized by degree of nutrition in cardiorespiratory fitness for both the boys and the girls. A post hoc Scheffé test was conducted for the variables where statistical significance had been established. The results indicated that both the male and female respondents with normal body weight had significantly higher values of cardiorespiratory fitness compared to those who were overweight or obese.

The individual correlation of variables for assessing obesity with cardiorespiratory fitness (CRF) was evaluated using Pearson’s correlation coefficient, as shown in Table 4. The results were examined for both genders. The obtained results demonstrated an inverse negative correlation of all variables for assessing obesity with cardiorespiratory fitness (CRF) in both genders. In the boys, a weak-to-moderate correlation was found, which was statistically significant for all variables, with the strongest correlation observed in the variable assessing the percentage of body fat (BF%). In the girls, the values were somewhat lower, and a significant correlation was not found for the variable assessing the waist-to-hip ratio (WHR).

Table 5 and Table 6 present the results of the regression analysis that determined the predictive values between variables for assessing obesity and cardiorespiratory fitness, respectively, for boys and girls. The results in Table 5, analyzing the boys, show the statistical significance of the set of predictor variables that are indicators of obesity on the criterion variable predicting cardiorespiratory fitness (CRF). The coefficient of determination (R^2^) explains nearly 22% of the common variability, while the remaining influence can be attributed to other abilities (motor, psychological, or others). Observing the beta regression coefficients, it is evident that the variable of body fat percentage (−0.60) has a statistically significant beta coefficient among boys. In Table 6, looking at the results for girls, statistical significance in the regression analysis can be observed. The coefficient of determination (R^2^) explains 19% of the common variability, which is somewhat lower than it is for boys. Statistically significant beta coefficients are found for the variables of body fat percentage (−0.98) and body mass index (−0.80) at a significance level of *p* < 0.05.

## 4. Discussion

The primary aim of this study was to investigate the relationship between cardiorespiratory fitness and obesity indicators in children aged between ten and eleven years. The main findings of this study show a significant correlation between obesity indicators and cardiorespiratory fitness in both boys and girls, with between 19% and 22% of the impact on functional ability attributed to obesity factors. This particularly applies to the variable assessing body fat percentage (BF%). This variable is significant for both genders and has the greatest impact as an obesity indicator on cardiorespiratory fitness for both genders. Regarding differences based on nutritional status, it is evident that children with normal body weight have significantly better cardiorespiratory fitness than those who are overweight or obese.

A large number of previous studies seek the most optimal way to quickly detect obesity, which, in combination with cardiorespiratory fitness (CRF), is the best indicator of health status in primary school children [17,32,41,42,43]. In our study, the results indicate that nearly 34% of the children were overweight or obese. In comparison to previous reearch [2], this percentage is somewhat lower among children in the Republic of Croatia. Additionally, a higher percentage of boys (38%) were classified as overweight or obese compared to girls (30%). This finding does not align with the normal occurrence in the growth and maturation process, where it has been established that girls have a consistently higher percentage of body fat than boys [44]. This fact is very interesting, and the reasons may likely be attributed to inadequate nutrition among children aged 10 or 11, considering that this has also become a public health issue in the Republic of Croatia.

Boys have a statistically significantly lower body fat percentage (BF%) than girls, which is biologically acceptable since, according to the laws of growth and development, women have a higher percentage of body fat than men [44]. Looking at the remaining indicators, it is evident that girls of this age have a significantly lower BMI, waist circumference, and hip circumference, as well as a lower waist-to-hip ratio (WHR). Additionally, girls have significantly lower values of waist-to-height ratio (WHtR) than boys. In the study by Bustos-Barahona [45], no gender differences were found in BMI, waist circumference (WC), WHtR, and BF%, while Füssenich (2016) and Dencker (2011) found a higher BF% in girls, consistent with our study [46,47]. According to Ostojić (2011), higher values of obesity indicators were found in girls compared to boys [48], while no differences were determined in relation to gender in BMI and WC [49,50,51]. When examining the variable for assessing cardiorespiratory fitness (CRF), evaluated through the 20-meter shuttle run, it is evident that boys completed significantly more laps than girls, suggesting that they also have a better-developed cardiorespiratory capacity and, consequently, a higher maximal oxygen uptake (VO_2_max) than girls. The results obtained align with those of previous studies [45,46,47,48,49,51,52]. However, the usual explanation for gender differences in CRF among children is based on a higher level of physical activity in boys, which is consistent with findings that boys are generally more physically active than girls [51].

When examining the differences in cardiorespiratory fitness (CRF) according to nutritional status, statistically significant differences were observed in both genders. Normal-weight boys had significantly better cardiorespiratory fitness (CRF) results than overweight or obese boys. The same results have been found in previous studies [23,49,50], while no differences were identified between overweight and obese boys. Identical results were obtained for girls, where normal-weight girls showed significantly better cardiorespiratory fitness (CRF) results compared to overweight or obese girls, with no differences found between overweight and obese girls. These results are consistent with research conducted on Brazilian children aged 7 to 14 years [23].

Our research showed that obesity indicators such as BF%, BMI, WHtR, and WC as a whole can affect the level of cardiorespiratory fitness (CRF) by about 22% in boys and 19% in girls. The highest individual statistically significant impacts on these values were related to BF% in boys, where the beta coefficient was (−0.60), and in girls, it was (−0.98). Similar results were found in research on Latin American children [53], which indicated that high percentages of BF, BMI, and WHtR were significantly associated with low CRF. Similar inverse relationships between CRF and body composition in youth exist in various settings, including the USA and Europe [54].

A negative correlation between obesity indicators and cardiorespiratory fitness (CRF) has also been found in previous studies [23,51,55]. A reduced level of BF% contributes to a higher level of CRF. Likewise, for the WHtR variable, it was evident that the values (β = −0.35) for girls and (β = −0.21) for boys contributed most to the impact of obesity indicators on CRF, although statistical significance was not established. Similar results were noted by Dencker (2011) [47]. Additionally, a significant positive correlation with CRF was found for BMI in girls. In this case, there is no meaningful explanation for why an increase in BMI would lead to an increase in CRF. This fact could only be attributed to growth and development, as students, especially girls, begin to grow rapidly at ages 10 and 11, leading to an increase in both their body height, body mass, and, consequently, BMI. According to Brand (2021), results indicate that there is no significant correlation between BMI and CRF in normally nourished boys, while such a correlation was established in normally nourished girls [23]. The BMI is the measure most commonly used to assess obesity, but it measures excess weight relative to height rather than body fat, and fat tissue is not evenly distributed throughout the body [56,57].

It has been established that higher physical fitness, or higher cardiorespiratory fitness (CRF), in primary school children is significantly associated with lower BMI, waist circumference (WC), and body fat percentage (BF%), which can be a significant predictor of a lower prevalence of metabolic syndrome later in life. As in our study, similar values have been obtained in previous research for boys, where higher CRF was associated with a lower BMI [58,59].

Physical activity should be more strongly promoted in health care systems, especially in school programs, as CRF is an indicator of regular physical activity, particularly aerobic exercises [60]. Therefore, CRF could be mandatorily applied in all health and educational systems in combination with anthropometric measures such as body fat percentage (BF%) and waist-to-height ratio (WHtR), which are potentially the strongest predictors of cardiorespiratory fitness [61]. The relationship between the cut-off values for low levels of CRF and obesity provides a clear insight into the connection between unfavorable cardiorespiratory fitness and lower health status, highlighting the need for public health interventions to increase physical fitness levels in the younger population. CRF should be an important measurement component from a young age to prevent general and abdominal obesity in the future [62]. Given that CRF is strongly associated with individual components of body composition, low levels of CRF in young individuals with high body fat may be a factor in the development of cardiovascular diseases in middle and older age [63].

Interventions that include physical exercise in young people have been shown to improve cardiorespiratory fitness (CRF) by an average of 30%, including children and adolescents with overweight and obesity [64]. The inadequate lifestyle of children and youth contributes to obesity, which is now one of the major health problems. A lack of participation in daily moderate-to-high-intensity physical activity is associated with poorer motor development in children [65]. For this reason, it is necessary to consider the rapid introduction of interventions in the school curriculum in the Republic of Croatia. Such interventions, for example, in a school environment, should promote physical activity to increase levels of CRF [66]. Physical education classes for younger school-aged students should be designed to create a motivational climate to encourage physical activity and creatively and successfully stimulate student participation in physical education classes [67]. The results of our research indicate that the percentage of body fat (BF%) is in a high and significant correlation with the number of laps run or higher CRF in students aged 10–11 years. It is essential to include the assessment of BF%, measured by bioelectrical impedance analysis (BIA), and CRF, measured by the 20-meter shuttle run test, in public health actions and school programs. Additionally, there is the possibility of using the waist-to-height ratio (WHtR) indicator to determine the state of cardiorespiratory capacity or obesity status in students in a more cost-effective manner when measurement instruments are lacking. WHtR has been shown to be a stronger predictor of CRF results compared to body mass index (BMI) and waist circumference (WC) [36]. Research suggests that measuring BF% via BIA and the manual measurement of WC can be interchangeable [68]. It is proposed that measuring WC, due to its simplicity and the ability to use the results to calculate WHtR, can be carried out during physical education classes and included in the core curriculum for this subject [36].

The strength of this study lies in establishing evidence of the connection between cardiorespiratory fitness (CRF) and relatively new indicators of obesity in the context of their predictive potential for CRF, which has been explored in several studies so far but has never been applied in the territory of the Republic of Croatia. Given that Croatia has a significant problem with the prevalence of obesity among young children, it is assumed that such research will help address this public health issue, not only based on the results obtained but also on the previous experiences of other countries.

The limitations of this study relate to the inability to generalize our findings due to the relatively small number of participants. It is necessary to use more data from a larger sample to confirm the results. Furthermore, all participants attended the same class and were almost the same age, so future research should include data from other age groups to validate our findings.

## 5. Conclusions

This research concludes that body fat percentage (BF%) is a significant and useful indicator of cardiorespiratory fitness among school-aged boys and girls. One of the recommendations of this study is the use of body composition analyzers in all schools in the Republic of Croatia, which would allow for the quick and efficient determination of body fat percentage (BF%) and nutritional status. Responsible educational policies in the Republic of Croatia should develop an action plan and intervention programs to combat obesity and overweight issues, which would also involve increasing physical activity among students, thereby improving cardiorespiratory fitness (CRF). Maintaining optimal body weight alongside high levels of CRF should be an important goal for promoting the health of children and adolescents.

## Figures and Tables

**Table 1 jfmk-09-00250-t001:** Descriptive indicators of anthropometric measurements and cardiorespiratory fitness of the subjects in this study.

Total (N = 423)						
Variable	M	SD	Min	Max	Skew	Kurt
Age	10.32	0.48	10.00	12.00	0.91	−0.83
BH (cm)	147.67	6.96	124.60	167.00	0.13	0.04
BW (kg)	41.78	10.44	21.90	82.60	0.86	0.62
BF (%)	20.38	7.83	5.30	44.40	0.38	−0.52
BMI (kg/m^2^)	18.97	3.70	12.90	31.40	0.82	0.25
WC (cm)	64.85	9.55	45.00	101.00	1.10	1.18
HC (cm)	81.13	8.99	64.00	110.00	0.57	−0.06
WHR (cm)	0.80	0.05	0.51	0.99	0.19	2.10
WHtR (cm)	0.44	0.06	0.29	0.64	1.02	0.92
Laps (frec)	23.93	14.29	3.00	76.00	1.29	1.37

M = mean; SD = standard deviation; Min = minimum values; Max = maximum values; Skew = skewness; Kurt = kurtosis; BH= body height; BW= body weight; BF% = body fat; BMI = body mass index; WC= waist circumference; HC = hip circumference; WHR = waist-to-hip ratio; WHtR = waist-to-height ratio; and Laps= laps in the 20 m shuttle test.

**Table 2 jfmk-09-00250-t002:** Results of the variance analysis between girls and boys.

	Boys (N = 211)		Girls (N = 212)		ANOVA	
Variable	M	SD	Mean	SD	F	*p*
Age	10.37	0.49	10.27	0.45	4.77	0.03
BH (cm)	147.42	6.80	147.93	7.13	0.57	0.45
BW (kg)	42.74	11.35	40.83	9.37	3.59	0.06
BF (%)	19.15	7.70	21.61	7.79	10.64	0.00 *
BMI (kg/m^2^)	19.43	4.05	18.51	3.26	6.61	0.01 *
WC (cm)	67.15	10.60	62.56	7.75	25.96	0.00 *
HC (cm)	82.15	9.59	80.11	8.24	5.54	0.02 *
WHR (cm)	0.82	0.06	0.78	0.05	46.21	0.00 *
WHtR (cm)	0.45	0.06	0.42	0.05	35.33	0.00 *
Laps (frec)	26.68	16.77	21.18	10.65	16.22	0.00 *

M = mean; SD = standard deviation; BH = body height; BW = body weight; BF% = body fat; BMI = body mass index; WC = waist circumference; HC = hip circumference; WHR = waist-to-hip ratio; WHtR = waist-to-height ratio; and Laps = laps in the 20 m shuttle test. * denotes significant differences, *p* < 0.05.

**Table 3 jfmk-09-00250-t003:** Results of the variance analysis between boys and girls classified according to the degree of nutrition.

Boys N = 211	Normal weight		Overweight		Obesity		ANOVA		Effect size
	N = 130		N = 51		N = 30				
	M	SD	M	SD	M	SD	F	*p*	η^2^
Number of Laps	31.95 *^a/b^	17.49	20.76	12.55	13.93	7.07	21.88	0.00	0.17
BF	14.24 *^a/b^	0.36	24.48 *^c^	0.57	31.34	0.74	275.08	0.00	0.73
BW	35.64 *^a/b^	0.51	49.65 *^c^	0.82	61.76	1.08	286.07	0.00	0.73
Girls N = 212	Normal weight		Overweight		Obesity		ANOVA		Effect size
	N = 149		N = 54		N = 9				
	M	SD	M	SD	M	SD	F	*p*	η^2^
Number of Laps	22.83 *^a/b^	11.49	17.74	6.86	14.56	7.57	6.69	0.00	0.06
BF	17.69 *^a/b^	0.38	29.65 *^c^	0.64	38.19	1.56	189.63	0.00	0.65
BW	36.27 *^a/b^	0.48	49.98 *^c^	0.79	61.36	1.94	167.70	0.00	0.62

M = mean; SD = standard deviation; F = value; ANOVA = analysis of variance; *p* < 0.05; * = *p*-value shows significant differences in Scheffe level; ^a^ = normal weight–overweight; ^b^ = normal weight–obesity; ^c^ = overweight–obesity; *p* < 0.05 denotes significant differences; BW = body weight; and BF = body fat.

**Table 4 jfmk-09-00250-t004:** The relationship between variables that are indicators of obesity and cardiorespiratory fitness.

Variable	Laps–CRF Boys	Laps–CRF Girls
BF	−0.46 *	−0.38 *
BMI	−0.41 *	−0.30 *
WC	−0.39 *	−0.26 *
WHR	−0.19 *	−0.07
WHtR	−0.39 *	−0.30 *

BF = body fat; BMI = body mass index; WC = waist circumference; WHR = waist-to-hip ratio; WHtR = waist-to-height ratio; Laps = laps in the 20 m shuttle test; and * = *p* < 0.05 denotes significant differences.

**Table 5 jfmk-09-00250-t005:** The results of the regression analysis for determining the predictive values between variables for assessing obesity and cardiorespiratory fitness in boys.

Number of Laps R = 0.47; R^2^ = 0.22; SE = 14.98; F = 11.59; and *p* = 0.00					
	*β*	SE	*B*	t Value (205)	*p*-Value
BF	−0.60	0.18	−1.31	−3.31	0.00 *
BMI	0.35	0.27	1.44	1.28	0.20
WC	−0.10	0.27	−0.15	−0.36	0.72
WHR	0.18	0.13	53.19	1.35	0.18
WHtR	−0.21	0.28	−56.34	−0.74	0.46

*β* = standardized beta coefficient; SE = standard error for beta coefficient; *B* = unstandardized beta coefficient; R = coefficient multiple correlation; R^2^ = coefficient of determination; F = test statistic; *p*-value for the significance of the whole model; BF = body fat; BMI = body mass index; WC = waist circumference; WHR = waist-to-hip ratio; WHtR = waist-to-height ratio; Laps= laps in the 20 m shuttle test; and * = *p* < 0.05 denotes significant differences.

**Table 6 jfmk-09-00250-t006:** The results of the regression analysis for determining the predictive values between variables for assessing obesity and cardiorespiratory fitness in girls.

Number of Laps R = 0.44; R^2^ = 0.19; SE = 9.64; F = 9.76; and *p* = 0.00					
	*β*	SE	*B*	t Value (206)	*p*-Value
BF	−0.98	0.22	−1.33	−4.36	0.00 *
BMI	0.80	0.30	2.61	2.69	0.01 *
WC (cm)	0.11	0.21	0.15	0.53	0.60
WHR	0.13	0.15	28.40	0.87	0.39
WHtR	−0.35	0.23	−79.62	−1.53	0.13

*β* = standardized beta coefficient; SE= standard error for beta coefficient; *B* = unstandardized beta coefficient; R = coefficient multiple correlation; R^2^ = coefficient of determination; F = test statistic; *p*-value for the significance of the whole model; BF% = body fat; BMI = body mass index; WC = waist circumference; WHR = waist-to-hip ratio; WHtR = waist-to-height ratio; Laps= laps in the 20 m shuttle test; *= *p* < 0.05 denotes significant differences.

## Data Availability

The data presented in this study are available on request from the corresponding author.

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
