# Peer review of "Indicators of Obesity and Cardiorespiratory Fitness in Croatian Children"

_jfmk, 2024, doi:10.3390/jfmk9040250_

Round 1
Reviewer 1 Report
Comments and Suggestions for Authors
In recent years, the issue of overweight and child obesity has been growing more acutely. The authors of the article within the scope of solving the issue of reducing the number of obese children in Croatia, have considered the relationship between indicators of body composition and functional fitness of children aged 10 and 11.
The manuscript given for review is characterized by a well-balanced structure of information presentation, the author describes the research in a logical and consistent manner, the research results are presented in an accessible manner, and the conclusions based on the obtained results are accordingly formed.
We especially would like to note the wide range of studies that has been analysed by the authors (68 reference sources), about a third of which are works published in the last five years. There are no signs of excessive self-citation, and the presented previous authors’ works allow us to trace their previously acquired experience.
The research design presented by the author allows us to fully follow its stages as well as allows us to test the research hypothesis. The description of the methods makes it possible to understand the general process of conducting the research.
The used images and tables allow us to visually present the obtained results and easily analyse them.
In addition to a positive impression of the article, you should pay attention to certain provisions that can improve the presented material:
- note to table 1 M=mean is not used and other designations should be the same, both in the note and in the table header (for example, Skew and SKEW), in table 2 it is worth indicating the units of indicators measurements;
- in point 2.3, please, provide information about the original source of this test (author, publication), and whether the use of this test is provided for the selected age category of the subjects;
- in point 2.4, information should be provided on whether differences in indicators between 10- and 11-year-old children have been clarified, in order to determine the validity of arranging them into one group;
- it is necessary to establish certain approach to the analysis of research results (table 1, 2) according to the degree of nutrition, due to the significant variability of such indicators as BW (kg), BF (%);
- table 3 needs technical correction;
- please, check the design of links and references [Cole et al., 2000];
- the author has not paid attention to the VO2 max indicator in the article when presenting the results, but in the discussion the author has provided these results (higher maximal oxygen uptake (VO2 max) than girls);
- the conclusions need to be revised regarding presenting specific results obtained during the research, as far as information presented in conclusions seems to be continuation of discussion.
Author Response
In recent years, the issue of overweight and child obesity has been growing more acutely. The authors of the article within the scope of solving the issue of reducing the number of obese children in Croatia, have considered the relationship between indicators of body composition and functional fitness of children aged 10 and 11.
The manuscript given for review is characterized by a well-balanced structure of information presentation, the author describes the research in a logical and consistent manner, the research results are presented in an accessible manner, and the conclusions based on the obtained results are accordingly formed.
We especially would like to note the wide range of studies that has been analysed by the authors (68 reference sources), about a third of which are works published in the last five years. There are no signs of excessive self-citation, and the presented previous authors’ works allow us to trace their previously acquired experience.
The research design presented by the author allows us to fully follow its stages as well as allows us to test the research hypothesis. The description of the methods makes it possible to understand the general process of conducting the research.
The used images and tables allow us to visually present the obtained results and easily analyse them.
In addition to a positive impression of the article, you should pay attention to certain provisions that can improve the presented material:
- note to table 1 M=mean is not used and other designations should be the same, both in the note and in the table header (for example, Skew and SKEW), in table 2 it is worth indicating the units of indicators measurements;
- in point 2.3, please, provide information about the original source of this test (author, publication), and whether the use of this test is provided for the selected age category of the subjects;
- in point 2.4, information should be provided on whether differences in indicators between 10- and 11-year-old children have been clarified, in order to determine the validity of arranging them into one group;
- it is necessary to establish certain approach to the analysis of research results (table 1, 2) according to the degree of nutrition, due to the significant variability of such indicators as BW (kg), BF (%);
- table 3 needs technical correction;
- please, check the design of links and references [Cole et al., 2000];
- the author has not paid attention to the VO2 max indicator in the article when presenting the results, but in the discussion the author has provided these results (higher maximal oxygen uptake (VO2 max) than girls);
- the conclusions need to be revised regarding presenting specific results obtained during the research, as far as information presented in conclusions seems to be continuation of discussion.

Reviewer 2 Report
Comments and Suggestions for Authors
Dear Authors,
you have conducted important research related to the epidemic of overweight and obesity among children and adolescents. The manuscript has been well developed in terms of science. Details comments:
1. Introduction – well developed. Only the last sentence “The findings…” is incomprehensible. What was the purpose of this sentence? I believe that ending the Introduction with only the purpose of the research is sufficient. You can possibly add a hypothesis.
2. The methods and research material have been well described.
3. The basic statistical analysis enriched with other parameters (e.g. kurtosis; skewness).
4. Results: Tables 1, Table 2, – no units of measurement, e.g. age, BMI. The analysis of the research results is correct.
5. Discussion – well conducted. The research results were referred to many research results of other authors.
6. The conclusions are supported by the obtained research results. However, the statement "Previous research has relied on BMI to determine obesity, which has been proven to be a very unreliable parameter", does not seem to be justified by the research results. Please justify this statement. Please even look at your research (Table 4, Table 5). In addition, until your research (methodology) is confirmed by several other researchers, there is no certainty that your conclusions can be accepted as truth that can be applied to the entire population.
7. References - please provide justification for each publication published before the year 2000. Usually in scientific studies we base ourselves on publications from the last 10-15 years.
8. References was developed contrary to the recommendations of the editors (see: Instructions for Authors). This section should be corrected.
9. Other: the text should be checked for style. There are errors, e.g. p. 3, section 2.2, 8th line from the bottom "based on" and "Based on" (repeat the same expression).
Author Response
- Introduction – well developed. Only the last sentence “The findings…” is incomprehensible. What was the purpose of this sentence? I believe that ending the Introduction with only the purpose of the research is sufficient. You can possibly add a hypothesis.
- The methods and research material have been well described.
- The basic statistical analysis enriched with other parameters (e.g. kurtosis; skewness).
- Results: Tables 1, Table 2, – no units of measurement, e.g. age, BMI. The analysis of the research results is correct.
- Discussion – well conducted. The research results were referred to many research results of other authors.
- The conclusions are supported by the obtained research results. However, the statement "Previous research has relied on BMI to determine obesity, which has been proven to be a very unreliable parameter", does not seem to be justified by the research results. Please justify this statement. Please even look at your research (Table 4, Table 5). In addition, until your research (methodology) is confirmed by several other researchers, there is no certainty that your conclusions can be accepted as truth that can be applied to the entire population.
- References - please provide justification for each publication published before the year 2000. Usually in scientific studies we base ourselves on publications from the last 10-15 years.
- References was developed contrary to the recommendations of the editors (see: Instructions for Authors). This section should be corrected.
- Other: the text should be checked for style. There are errors, e.g. p. 3, section 2.2, 8th line from the bottom "based on" and "Based on" (repeat the same expression).
